# The Isoelectric Point of an Exotic Oxide: Tellurium (IV) Oxide

**DOI:** 10.3390/molecules26113136

**Published:** 2021-05-24

**Authors:** Marek Kosmulski, Edward Mączka

**Affiliations:** Laboratory of Electrochemistry, Lublin University of Technology, PL-20618 Lublin, Poland; e.maczka@pollub.pl

**Keywords:** electric double layer, point of zero charge, isoelectric point, zeta potential, specific adsorption

## Abstract

The pH-dependent surface charging of tellurium (IV) oxide has been studied. The isoelectric point (IEP) of tellurium (IV) oxide was determined by microelectrophoresis in various 1-1 electrolytes over a concentration range of 0.001–0.1 M. In all electrolytes studied and irrespective of their concentration the zeta potential of TeO_2_ was negative over the pH range 3–12. In other words the IEP of TeO_2_ is at pH below 3 (if any). TeO_2_ specifically adsorbs ionic surfactants, and their presence strongly affects the zeta potential. In contrast the effect of multivalent inorganic ions on the zeta potential of TeO_2_ is rather insignificant (no shift in the IEP). In this respect TeO_2_ is very different from metal oxides.

## 1. Introduction

Tellurium dioxide is a white crystalline compound, chemically inert and practically insoluble in water. It occurs in Nature, and it can be easily obtained, for example, by synthesis from the constituent elements. TeO_2_ is used as an acousto-optic material and in optical fibers [1,2]. TeO_2_ is not particularly precious: a price for 1 kg of elementary tellurium is about $80.

Scientific knowledge on adsorption properties of TeO_2_ is practically absent. A search for adsorption AND (tellurium oxide OR tellurium dioxide) in scientific databases results in a dozen papers, which describe some kind of adsorption and they refer somehow to TeO_2_, but not to adsorption on TeO_2_ from solution. Very few papers describe adsorption of gases on TeO_2_ in the context of its possible application as a gas sensor [3].

A few papers report zeta potential of TeO_2_ which is directly related to adsorption of ions from solution. TeO_2_ nanoparticles had a zeta potential of about −50 mV at pH 5.3 (ionic strength not specified) [4]. Those particles were prepared in the presence of organic acids which might have induced negative zeta potentials. Properties of such particles are not relevant to electrokinetic properties of pure TeO_2_ in the presence of inert electrolyte. Another specimen of TeO_2_ had a zeta potential of +29 mV in deionized water [5]. The pH was not reported which makes this result useless in determination of the isoelectric point, IEP. Also, the level of purity of TeO_2_ used in electrophoretic measurements was not discussed.

The chemical analogs of TeO_2_, that is, SO_2_ and SeO_2_ (dioxides of the 16th group elements) undergo reactive dissolution in water, and PoO_2_ is seldom studied because of its high radioactivity, and high cost. Therefore, TeO_2_ is the only dioxide of the 16th group element which can be considered as adsorbent of ions from solution. We hypothesize that TeO_2_ behaves like the other sparingly soluble oxides [6,7], that is, it shows pH-dependent surface charging caused by reversible pH-dependent protonation and deprotonation of the surface oxygen atoms according to reactions (1) and (2):≡TeOH + H^+^ = ≡TeOH_2_^+^ (below PZC, positively charged surface) (1)
≡TeOH = ≡TeO^-^ + H^+^ (above PZC, negatively charged surface) (2)
where ≡Te is a tellurium atom on the surface.

We further hypothesize that positively charged surfaces of TeO_2_ adsorb anions by Coulombic attraction, and negatively charged surfaces of TeO_2_ adsorb cations by Coulombic attraction. The sign of the surface charge depends on pH, and more precisely on the relationship between the actual pH and the point of zero charge (PZC). Isoelectric point (IEP) and PZC are basically two different quantities, but for metal oxides in the presence of 1-1 salts (1st group metal halides, nitrates (V) and chlorates (VII)), they usually occur at the same pH value. We hypothesize that also with TeO_2_, the IEP determined by electrokinetic measurements in 1-1 salts coincides with the PZC. We further hypothesize that on top of the aforementioned electrostatic adsorption certain ions, e.g., ionic surfactants will specifically adsorb on TeO_2_, that is, these ions can adsorb even against electrostatic repulsion. In order to predict the sign of surface charge of we need to know the PZC. Unfortunately, there are no scientific papers reporting PZC/IEP of TeO_2_. We may estimate PZC of TeO_2_ using the correlations between points of zero charge and well-established physical properties, e.g., ionic radii and electronegativities [6,7] (Equations (3)–(10)). We cannot use a correlation between PZC and the 1st hydrolysis constant of metal ions, because the Te^4+^ species in aqueous media is non-existing:PZC = 14.9 − 2.19·valence (not weighted)(3)
PZC = 15 − 2.26·valence (weighted) (4)
PZC = 17.74 − 1.36·electronegativity of oxide (not weighted) (5)
PZC = 17.88 − 1.4·electronegativity of oxide (weighted) (6)
PZC = 12.4 − 0.88·*z/r* (not weighted) (7)
PZC = 11.66 − 0.75·*z/r* (weighted) (8)
PZC = 15.03 − 7.73·*z/R*· (not weighted) (9)
PZC = 15.28 − 8.08*·z/R* (weighted) (10)

The PZC of TeO_2_ estimated from the Equations (3)–(10) [7] are summarized in Table 1.

In Table 1 the electronegativity of an oxide is the electronegativity of metal plus 1/2 the metal valence times the electronegativity of oxygen, *z* is the valence of metal in the oxide, *r* is the radius of metal cation and *R* is *r* plus two radii of oxygen anion. “Weighted” is the PZC estimated from weighted PZC of other metal oxides, that is PZC of a certain oxide is taken with a weight proportional to the logarithm of the number of literature references reporting the PZC of this oxide. Particular correlations lead to very different PZC of TeO_2_ ranging from 5.3 to 8.5.

The PZC of oxides is obtained experimentally as a common intersection point CIP of at least three surface charging curves for different concentrations of a 1-1 electrolyte. This method requires a high specific surface area (>>100 m^2^/L), otherwise the potentiometric titration curves of the dispersions do not differ much from the titration curves of the electrolyte. The condition recalled above (>>100 m^2^/L) is a rule of thumb rather than a strict principle, and the applicability of salt titration depends also on other factors than the solid-to-liquid ratio. Even with relatively high specific surface area, many materials fail to produce a sharp CIP, but instead the titration curves obtained at various ionic strength merge over a broad pH range. This is (usually) the case for silica but merging titration curves have been also reported for other oxides [7]. A set of titration curves without CIP is useless in determination of the PZC. Inert electrolyte titration may be used to determine the PZC in the systems where the classical potentiometric titration fails [8]. PZC obtained by titration is affected by specific adsorption of ions, also by impurities in the studied material. CIP is often observed in impure materials, but such a CIP is different from the pristine PZC [7]. Discrepancies between PZC reported in different publications for the same oxide are chiefly due to impurities. Therefore, PZC obtained by titration has to be confirmed by electrokinetic measurements. Matching IEP and CIP are obtained in pure oxides while in impure materials, the CIP and IEP do not match. Electrokinetic measurements have also been used as a standalone method of determination of the PZC of oxides (which is assumed to match the IEP), but such a method is uncertain because of high sensitivity of IEP to impurities.

Production of materials with high specific surface area is not trivial, and commercially available specimens of high-purity TeO_2_ consist of coarse particles, and they are not suitable for determination of PZC by titration. Several recipes for TeO_2_ nanoparticles have been reported [9,10,11]. Due to a high specific surface area such nanomaterials may be more suitable for studies of the PZC by titration than coarse particles available commercially. The problem with nanoparticles is about their purity. The experience with other oxides shows that many nanomaterials contain surface-active impurities which shift the PZC. Therefore in order to test our hypothesis we used coarse particles of high-purity TeO_2_ rather than fine particles of unknown level of purity.

## 2. Results and Discussion

### 2.1. BET and XRD

The XRD pattern is shown in Figure 1a, and it represents tellurite, β-TeO_2_. Similar positions of peaks in XRD pattern of tellurite have been reported by others [4,12,13]. Narrow peaks indicate large crystallites in our specimen.

The BET specific surface area of TeO_2_ was 0.27 m^2^/g. This figure corresponds to monodispersed spherical particles 4 μm in diameter, which is the upper limit of particle size suitable for microelectrophoretic measurements. However, the actual particles are neither monodispersed nor spherical (Figure 1b), and on top of large particles (>10 µm) there is a substantial fraction of submicron particles in the sample. The size distribution is very broad and we find sufficient number of submicron particles to carry out an electrophoretic measurement. We also believe that the particle radii obtained by DLS (vide infra) which were 200–400 nm when the zeta potential was >50 mV in absolute value represent the actual sizes of particles whose zeta potentials were measured while the larger particles settled down, and they were not transferred into the zetameter cell. The smallest particles shown in Figure 1b roughly match the aforementioned sizes obtained by DLS.

### 2.2. Zeta Potential and Particle Size

The zeta potential of TeO_2_ in various 1-1 electrolytes is shown in Figure 2, Figure 3, Figure 4, Figure 5, Figure 6 and Figure 7. These results can be summarized as follows. The absolute values of zeta potentials at constant pH decrease as the ionic strength increases. This behavior is commonplace in metal oxides [7], and it can be explained in terms of various models of the electric double layer. The maximum absolute value of the zeta potential of TeO_2_ is about 50 mV in 0.001 M electrolyte, 30 mV in 0.01 M electrolyte, and 15 mV in 0.1 M electrolyte. These absolute values are lower than typical results obtained with other oxides (in colloidal silica −100 mV in 0.001 M electrolyte, −60 mV in 0.01 M electrolyte, and −40 mV in 0.1 M electrolyte are commonplace), but a few publications report maximum absolute values of the zeta potential similar as the values obtained in this study [7]. The absolute value of the zeta potential at given pH decreases as the salt concentration increases according to a commonly observed trend. There was no specific salt effect (NaCl vs. KCl, etc.) at salt concentrations of 0.001 and 0.01 M.

The zeta potentials are negative at pH > 3. At pH 2–3 we have a mixture of small positive and small negative values. This result suggests an IEP between pH 2 and 3, that is, much lower than the predicted values in Table 1. Interestingly enough with 0.1 M electrolyte the zeta potentials assumed only negative values irrespective of the nature of the salt. The difficulties with exact determination of the IEP from electrokinetic curves are commonplace, e.g., in spite of large body of experimental data the position of IEP of silica is still under debate. The results presented in Figure 2, Figure 3, Figure 4, Figure 5, Figure 6 and Figure 7 are in line with high negative zeta potentials at pH 5.3 reported in [4], although such a result could also be due to specific adsorption of acetate and gallate anions. There is no systematic effect of the nature or concentration of electrolyte on the apparent IEP: with Li, Na and K, and with chloride, perchlorate and nitrate we have a few electrokinetic curves with only negative zeta potentials and a few other electrokinetic curves with positive zeta potentials in the pH range 2–3. Apparently, chlorides, nitrates V and chlorates VII of alkali metals are inert electrolytes for TeO_2_. Inert character of these salts has been confirmed for many other oxides [7].

The electrokinetic properties of TeO_2_ in the presence of LiClO_4_ are different from other 1-1 salts, namely, in 0.05 and 0.1 M LiClO_4_ the zeta potential was clearly negative even at pH as low as 2, while in 0.1 M solutions of other 1-1 salts it was close to zero over the pH range 2–3. This effect may be due to specific adsorption of ions (in this case of anions) from concentrated solutions of electrolytes, which are indifferent at low concentrations [14]. The shifts in the IEP of metal oxides in the presence of 1-1 electrolytes were observed at electrolyte concentrations >1 M, but with silica such shifts were even observed at electrolyte concentrations below 0.1 M [15]. The other explanation of unusual behavior of TeO_2_ in the presence of LiClO_4_ is that both TeO_2_ and perchlorate anion are redox-active in acidic medium.

The results presented in Figure 2, Figure 3, Figure 4, Figure 5, Figure 6 and Figure 7. are more scattered than most electrokinetic curves presented in the literature. The size of the TeO_2_ particles at the upper limit of particle size suitable for microelectrophoretic particles combined with high specific density explains the difficulties in the measurements. The specific density of our TeO_2_ is 5.67 g/cm^3^. We attempted to grind the material by means of a vibratory mill (mortar and ball), but our attempt was unsuccessful.

We also studied the zeta potentials of TeO_2_ in the presence of ionic surfactants (Figure 8). The zeta potential in the presence of sodium dodecylsulfate SDS was about −60 mV and it was pH-independent. Also the increase in SDS concentration from 10^−4^ to 10^−3^ M had rather insignificant effect on the zeta potential. This result is due to high affinity of SDS to the surface combined with low specific surface area of the powder. Thus, the uptake of SDS anions by TeO_2_ surface reaches a maximum value at relatively low SDS concentration, and the negative charge of pre-adsorbed SDS anions prevents from further uptake of these anions. The zeta potential in the presence of cetyltrimethlyammonium bromide CTMABr was about +50 mV and it was pH-independent at pH > 3. Also the increase in CTMABr concentration from 10^−4^ to 10^−3^ M had rather insignificant effect on the zeta potential. Lower absolute value of zeta potential in the presence of CTMABr as compared with SDS may be due to the fact that the negative charge of adsorbed SDS anions and the negative pristine surface charge of TeO_2_ add up, while the positive charge of adsorbed CTMA cations is partially balanced out by the negative pristine surface charge of TeO_2_. Unlike with SDS, the adsorption of CTMA cations on TeO_2_ is affected by the electrostatics. At pH about 2.5 TeO_2_ is not charged, and the effect of CTMABr on its zeta potential is insignificant. CTMABr adsorbs on the negatively charged surface, but not on electroneutral or positively charged surface. In this respect TeO_2_ is very different from metal oxides, which strongly adsorb CTMA cations even at pH below their pristine PZC [16].

We also studied the zeta potentials of TeO_2_ in the presence of phosphate (Figure 9). The zeta potential was practically unaffected by the presence of phosphate (up to 10^−3^ M). In this respect TeO_2_ is very different from metal oxides, which strongly adsorb phosphate and other multivalent inorganic anions, and their IEP is shifted to low pH even at low concentrations of these anions [17]. In contrast with metal oxides, the effect of inorganic anions on the zeta potential of silica was seldom studied. A few results compiled in ref. [18] show that the uptake of inorganic anions by silica is low, and their effect on its zeta potential is rather insignificant.

We also studied the zeta potentials of TeO_2_ in the presence of barium (Figure 10). The IEP was practically unaffected by the presence of barium (up to 10^−3^ M). In this respect TeO_2_ is very different from metal oxides, which strongly adsorb barium and other multivalent inorganic cations, and their IEP is shifted to high pH even at low concentrations of these cations [19,20]. Depression of negative zeta potentials in 10^−3^ M Ba(NO_3_)_2_ at pH > 7 with respect to 10^−3^ M NaCl can be interpreted as the effect of increase in the ionic strength and it does not unequivocally indicate specific adsorption of Ba. In other words we do not need to invoke specific adsorption to explain the difference between Ba(NO_3_)_2_ (Figure 10) and NaH_2_PO_4_ (Figure 9): in view of divalent counterion (Figure 10) and monovalent counterion (Figure 9), the difference may be due to pure electrostatics. Moreover, NaH_2_PO_4_ is a weak electrolyte.

The particle radii in the dispersions of TeO_2_ were very scattered, but they showed a common pattern. In 0.1 M 1-1 electrolyte the particle radius was >1000 nm. In 0.01 and 0.001 M 1-1 electrolyte the particle radius was >1000 nm at pH < 4, when the negative zeta potential was low in absolute value, and 200–1000 nm at pH > 4, when the negative zeta potential was high in absolute value. This result is in line with a typical behavior of colloidal particles which show high particle size when the zeta potential (negative or positive) is low in absolute value due to coagulation. In the presence of ionic surfactants the particle radius was 200–400 nm irrespective of the pH. Again, the particle size was low when the zeta potential was high in absolute value.

### 2.3. Salt Titration

The results of salt titration of TeO_2_ with KCl at 25 °C are presented in Figure 11. The point where KCl addition does not affect the pH falls at pH 6.4 which is halfway between the PZC predicted from a correlation with the valence and from the correlation with *z*/*R* (Table 1). Relatively high apparent PZC is also in line with positive zeta potential at (probably) nearly neutral pH reported in [5]. The apparent PZC obtained by salt titration (Figure 11) is much higher than the IEP determined by electrophoresis (Figure 2, Figure 3, Figure 4, Figure 5, Figure 6 and Figure 7). Discrepancies between CIP and IEP even in apparently high-purity powders are commonplace in the literature [7]. They have been interpreted in terms of the presence of specifically adsorbing ions as an impurity in a high-purity reagent. Even a very small mass fraction of specifically adsorbing ions can induce a substantial shift in the IEP. Yet in the present case we believe that our IEP (pH < 3 if any) rather than the zero-point obtained by salt titration (pH 6.4) represents the actual pristine PZC. This is because our salt titration experiment was carried out beyond the normal application range of this method. We only have 54 m^2^/L of the surface area of solid in our experiment and the salt titration method requires >> 100 m^2^/L. The electrokinetic experiment directly indicates the sign of the particle charge while the salt titration method is based on several assumptions. The acid-base reactions in dispersions of metal oxides include surface protonation-deprotonation as well as protonation-deprotonation of solution monomers which are the products of dissolution of sparingly soluble metal oxides. The acid-base reactivity of solution monomers can be neglected when the solubility of oxide is very low and the surface area is very high, but apparently this not the case in our experiment. The apparent zero point obtained by salt titration decreased when the solid-to-liquid ratio increased, and this is a strong argument against using the apparent zero point from Figure 11 as a material constant. Perhaps some kind of extrapolation of apparent PZC obtained by salt titration to infinite solid-to-liquid ratio would result in the real PZC. Unfortunately, direct experiments at sufficiently high solid-to-liquid ratio cannot be conducted due to high viscosity of very concentrated dispersions. In our previous studies we applied the salt titration method to powders having much higher specific surface area (at least 10 m^2^/g), and such problems were not encountered. The results presented in Figure 11 are an example of an (expected) failure: the apparent PZC was dependent on the solid-to-liquid ratio. This is why we only performed such measurements with one salt (KCl, different solid-to-liquid ratios) and we have not attempted similar measurements with other salts. Moreover, the salt-specific surface-charging behavior is observed at high salt concentrations [14] while the salt titration is only efficient at low total salt concentration (addition of salt to dispersions, in which the salt concentration is already high has rather insignificant effect on pH, even far from the PZC).

We also determined the natural pH of dispersion composed of 10 g of TeO_2_ and 50 mL of water, which is 4.81. Although the present authors are critical about using the natural pH of concentrated dispersion to determine the PZC, many other authors identified the natural pH of concentrated dispersion with the PZC [7].

### 2.4. Solubility

In the older literature (cf. [18] for details) we found allegations that the PZC of metal oxides matches the pH of minimum solubility. The present authors do not recommend pH of minimum solubility as a method of determination of PZC, but we can easily check how it works with TeO_2_, because the solubility data are available [21].

According to Figure 1 in [21] the minimum of solubility falls at pH 5.3, which coincides with the lowest predicted PZC in Table 1. The minimum solubility is 10^−7^ M (total concentration of soluble Te (IV) species), and it increases to 10^−4^ M at pH 2, and to 10^−1^ M at pH 10. Reference [21] presents also a Pourbaix-type plot for Te. Within the electrochemical window of water, elementary Te, Te(IV) and Te(VI) can be thermodynamically stable over various ranges of pH and redox potential.

## 3. Materials and Methods

TeO_2_ (99.995%) from Aldrich (St. Louis, MO, USA) was used as obtained. The other chemicals were of analytical grade from POCh (Lublin, Poland). Electrophoretic mobility and particle size were measured by means of a Zetasizer (Malvern, Malvern, UK). Zeta potential was estimated by the Smoluchowski equation. The measurements were carried out in fresh dispersions (a few hours after preparation) at 25 °C. The dispersions containing specifically adsorbing ions were also 0.001 M in NaCl. While in our previous electrokinetic study with Fe(OH)_2_ we designed a special procedure to exclude O_2_ and CO_2_ and in our previous electrokinetic study with BeO we designed a special procedure to exclude CO_2_ and no special efforts were made to exclude O_2_ or CO_2_ in the current electrokinetic study with TeO_2_. In acidic solution, the solubility of CO_2_ is low. Moreover, we are not aware of any special affinity of TeO_2_ to O_2_ or CO_2_. Exclusion of O_2_ or CO_2_ is not necessary in electrokinetic studies of most metal oxides, and the IEP obtained with and without exclusion of O_2_ or CO_2_ are often identical [6,7]. We also conducted a literature search on possible carbonate complexes of inorganic tellurium, but apparently such complexes do not exist or at least have not been detected yet. Negative charge of TeO_2_ particles is also an argument against hypothetical adsorption of carbonate anions and their hypothetical effect on the zeta potential.

An Empyrean system from PANalytical (currently Malvern, Malvern, UK) was used to obtain the XRD pattern. Gemini V from Micromeritics (Norcross, GA, USA) was used to measure the specific surface area. Adsorption of nitrogen at its boiling point was studied. In view of low specific surface area we used relatively large samples of powder (about 1 g) in specific surface area measurements, and we repeated the measurement 3 times. The samples were outgassed at 300 °C for 1 h. The data points for *p*/*p*_0_ < 0.3 were used to calculate the specific surface area from the linearized form of BET equation.

Salt titration of dispersion of 10 g of TeO_2_ in 50 mL of water was carried out at 25 °C in nitrogen atmosphere. 50 μL portions of 3 M KCl were added to pre-equilibrated dispersion, and the change in the pH induced by salt addition was recorded. When the pH increased, 0.1 or 0.005 M KOH was added to further increase the pH. When the pH decreased, 0.01 M HCl was added to further decrease the pH. After each addition of salt, acid or base the dispersion was equilibrated. The initial dispersion was stirred in a stream of nitrogen for about 1 h before the titration to remove any trace of CO_2_ and O_2_ from the dispersion, and an overpressure prevented the flow of external air into the reactor. The salt titration relies on very small changes in pH (cf. Figure 11) so any trace of CO_2_ in the reactor may affect the results, especially at pH about 7. In this respect the salt titration is very different from electrophoresis where 0.01 pH unit or so is immaterial (cf. the sizes of the symbols in Figure 2, Figure 3, Figure 4, Figure 5, Figure 6, Figure 7, Figure 8, Figure 9 and Figure 10 which are about 0.1 pH unit).

The pH measurements were carried out using a combined electrode calibrated against 3 commercial, fresh pH-buffers (pH 4, 7 and 10, POCh). The pH was measured just before the injection of the dispersion into the zetameter cell. The time of the contact between the electrode (glass!) and the dispersion was minimized to avoid silica contamination.

## 4. Conclusions

We showed that the adsorption properties of TeO_2_ are only partially similar to those of metal oxides. TeO_2_ shows a pH-dependent surface charging, and its zeta potential is strongly influenced by the adsorption of ionic surfactants. On the other hand, multivalent inorganic ions have rather insignificant effect on the zeta potential of TeO_2_. The MUSIC model [22] offers a possibility of prediction of PZC of oxides from the crystallographic data. So far it has been used for oxides whose PZC is already well-documented, but it can also be used for oxides whose PZC is less well-documented, and this may be a suggestion for further research.

TeO_2_ is one of numerous common materials whose adsorption properties are almost unknown. Obviously TeO_2_ will not compete with silica or with activated carbon as an adsorbent for practical applications, but perhaps adsorption studies are too focused on a small number of adsorbents (classes of adsorbents) whose adsorption properties are already well-documented while the adsorption properties of many other common materials remain unknown. Even if the studies of adsorption on materials like TeO_2_ have limited practical meaning, they may be important as fundamental research which will help to better understand the adsorption phenomenon.

## Figures and Tables

**Figure 1 molecules-26-03136-f001:**
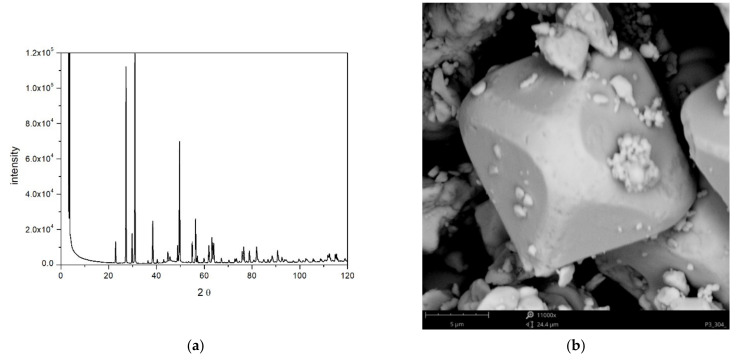
(**a**) The XRD pattern. (**b**) Photograph of TeO_2_.

**Figure 2 molecules-26-03136-f002:**
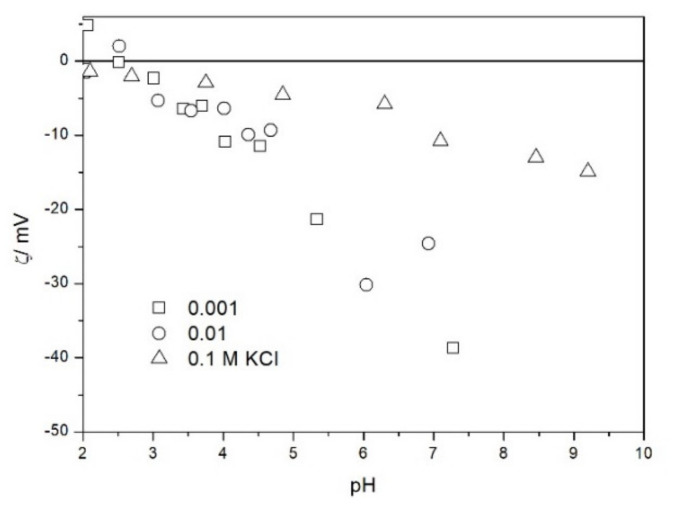
The zeta potential of TeO_2_ in KCl.

**Figure 3 molecules-26-03136-f003:**
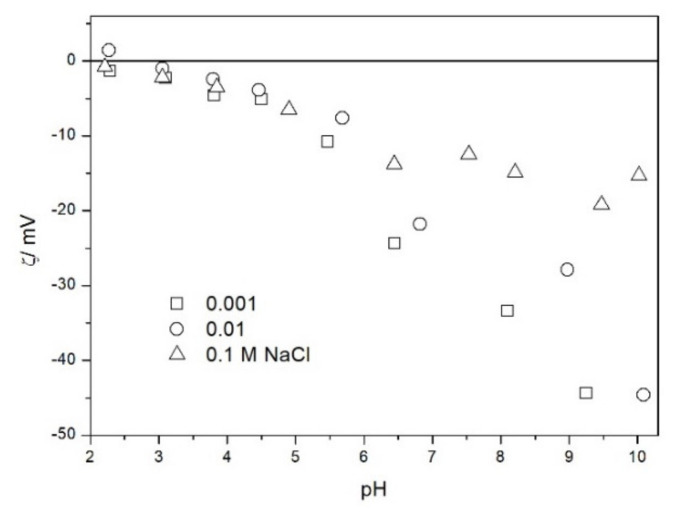
The zeta potential of TeO_2_ in NaCl.

**Figure 4 molecules-26-03136-f004:**
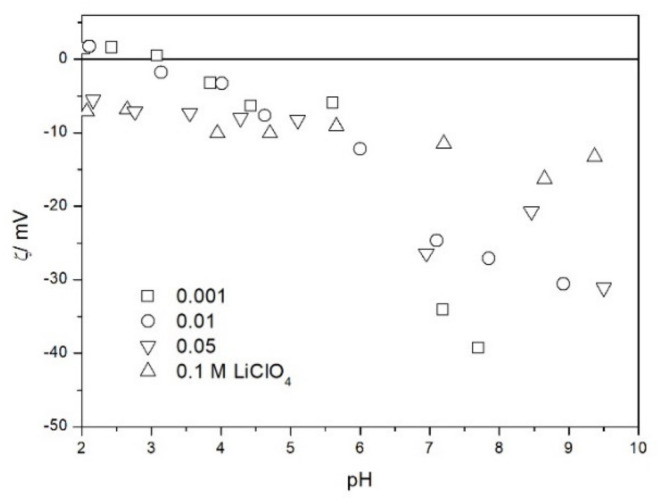
The zeta potential of TeO_2_ in LiClO_4_.

**Figure 5 molecules-26-03136-f005:**
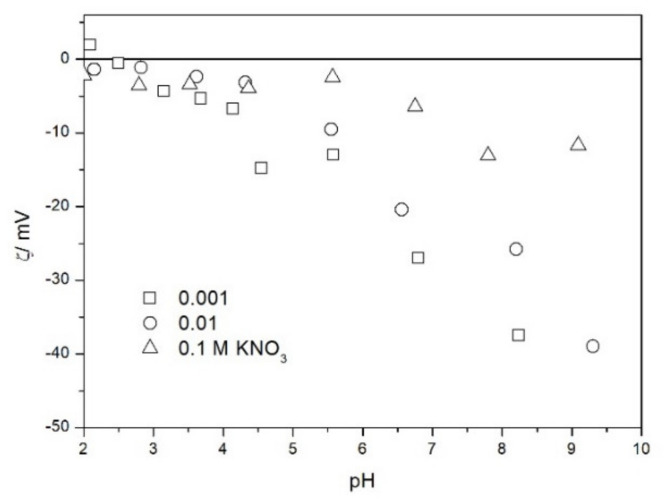
The zeta potential of TeO_2_ in KNO_3_.

**Figure 6 molecules-26-03136-f006:**
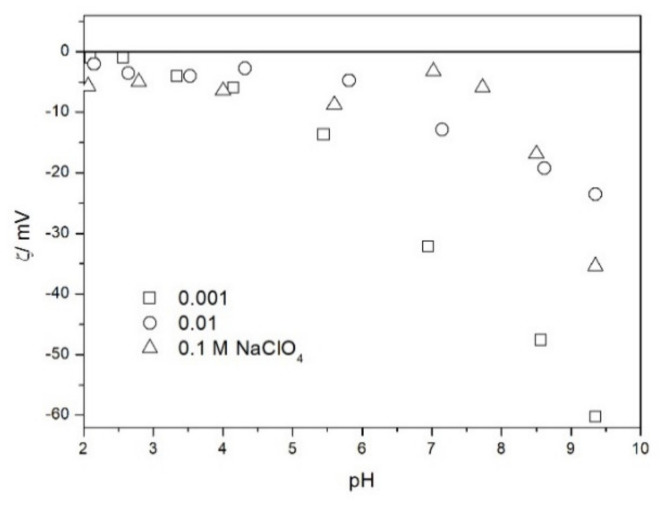
The zeta potential of TeO_2_ in NaClO_4_.

**Figure 7 molecules-26-03136-f007:**
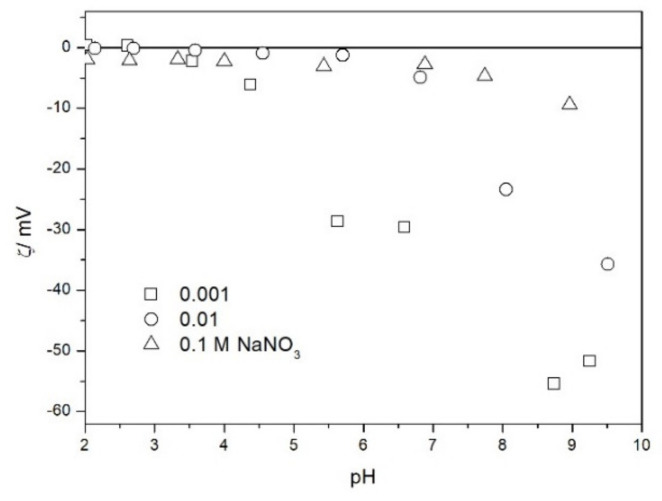
The zeta potential of TeO_2_ in NaNO_3_.

**Figure 8 molecules-26-03136-f008:**
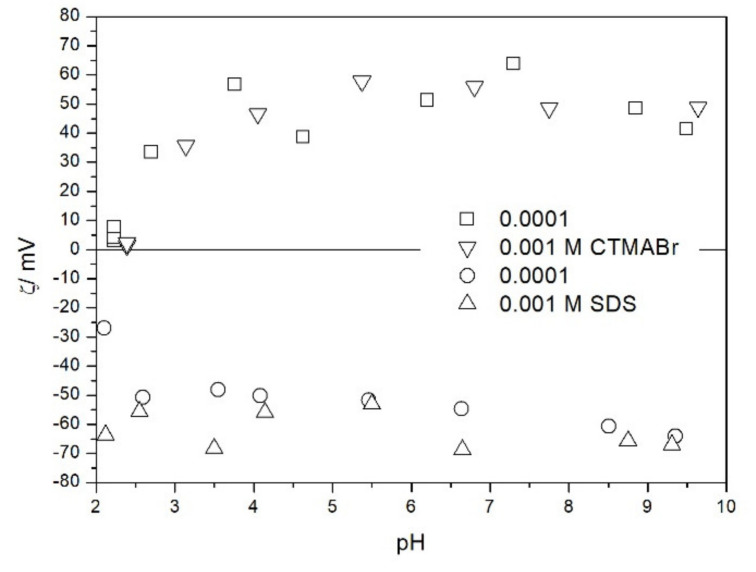
The zeta potential of TeO_2_ in the presence of ionic surfactants and 0.001 M NaCl.

**Figure 9 molecules-26-03136-f009:**
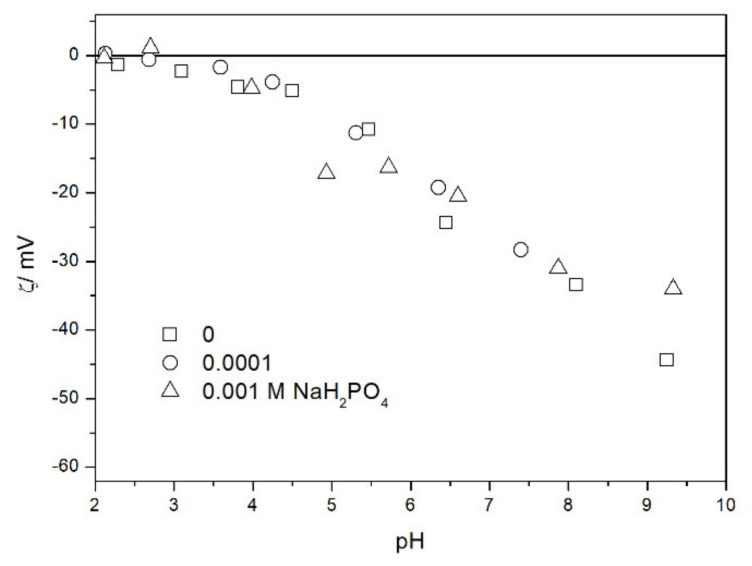
The zeta potential of TeO_2_ in the presence of phosphate and 0.001 M NaCl.

**Figure 10 molecules-26-03136-f010:**
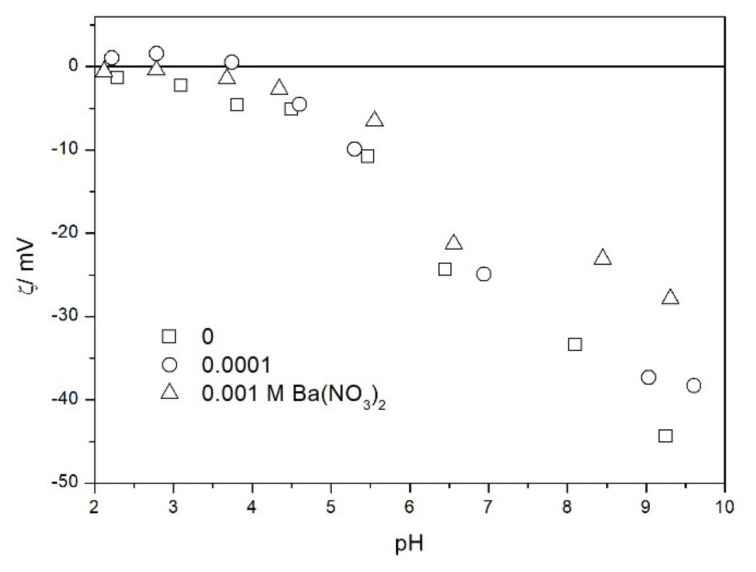
The zeta potential of TeO_2_ in the presence of barium and 0.001 M NaCl.

**Figure 11 molecules-26-03136-f011:**
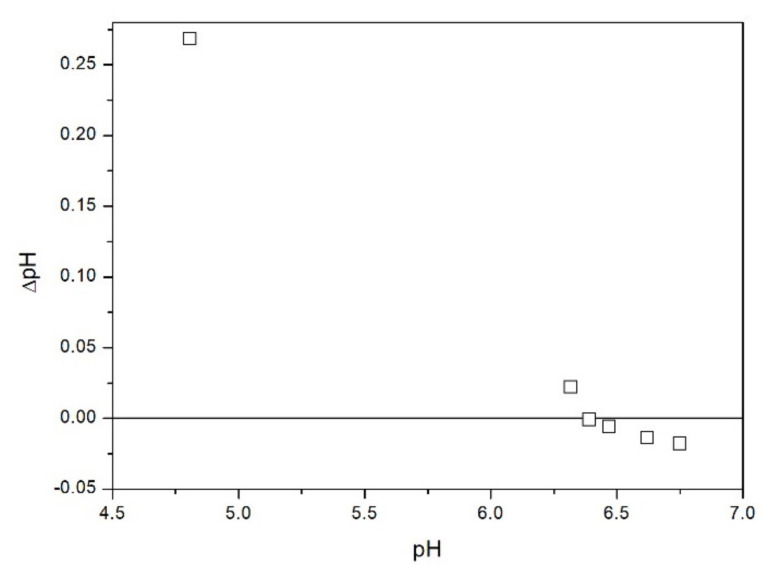
Shifts in the pH of TeO_2_ dispersion induced by addition of KCl.

**Table 1 molecules-26-03136-t001:** PZC of TeO_2_ estimated from correlations reported in [7].

Quantity	Not Weighted	Weighted
valence	6.20	6.00
electronegativity of oxide	5.54	5.33
*z/r*	8.51	8.57
*z/R*	6.80	6.70

## Data Availability

The data presented in this study are available on request from the corresponding author. The data are not publicly available. All significant data related to this research are presented in the paper.

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
