# Peer review of "The Isoelectric Point of an Exotic Oxide: Tellurium (IV) Oxide"

_molecules, 2021, doi:10.3390/molecules26113136_

Round 1

Reviewer 1 Report

The manuscript entitled "IEP of Exotic Oxides: Tellurium (IV) Oxide" represent a rich set of experimental data aimed to characterize the IEP of TeO2 and to point out new possibilities for the application of this relatively cheap inert material. The supervisor of the study is a leading expert in the field of surface properties and EDL of oxides, so the results are clearly presented and analyzed. I would be glad to see more comments on the relation to other surface properties, e.g. wetting by different liquids, possible inclusion in dispersion systems for new materials, etc. I suppose that these will be discussed in forthcoming studies. 

The presented manuscript is well prepared and might be published after minor revisions.

- Additional discussion is necessary to compare the results with NaH2PO4 and Ba(NO3)2 - why 1 mM of the phosphate do not change the ionic strength while the barium salt change it at pH above 7?  

- The authors could add, if they find appropriate, an image of the particles for the readers to get better idea for the sizes and shapes that are discussed. Micrometer particles are well visualized by an optical microscope.

- As for the big particles possibly disturbing the measurements they could be easily removed by a slight centrifugation for example.

Author Response

The manuscript entitled "IEP of Exotic Oxides: Tellurium (IV) Oxide" represent a rich set of experimental data aimed to characterize the IEP of TeO2 and to point out new possibilities for the application of this relatively cheap inert material. The supervisor of the study is a leading expert in the field of surface properties and EDL of oxides, so the results are clearly presented and analyzed. I would be glad to see more comments on the relation to other surface properties, e.g. wetting by different liquids, possible inclusion in dispersion systems for new materials, etc. I suppose that these will be discussed in forthcoming studies.

The presented manuscript is well prepared and might be published after minor revisions.

- Additional discussion is necessary to compare the results with NaH2PO4 and Ba(NO3)2 - why 1 mM of the phosphate do not change the ionic strength while the barium salt change it at pH above 7? 

The following phrase was added

Depression of negative zeta potentials in 10-3 M Ba(NO3)2 at pH>7 with respect to 10-3 M NaCl can be interpreted as the effect of increase in the ionic strength and it does not unequivocally indicate specific adsorption of Ba. In other words we do not need to invoke specific adsorption to explain the difference between Ba(NO3)2  (Fig. 10) and NaH2PO4 (Fig. 9): in view of divalent counterion (Fig. 10) and monovalent  counterion (Fig. 9), the difference may be due to pure electrostatics. Moreover NaH2PO4 is a weak electrolyte.

- The authors could add, if they find appropriate, an image of the particles for the readers to get better idea for the sizes and shapes that are discussed. Micrometer particles are well visualized by an optical microscope.

We added a photo.

- As for the big particles possibly disturbing the measurements they could be easily removed by a slight centrifugation for example.

Actually we did not have to do anything special to remove them: they just settled down, as explained in the revised text.

Reviewer 2 Report

Review for molecules-1220561

This is an important paper dealing with surface properties of an oxide that has previously been little explored. I wonder why it is not submitted to a colloid/interface journal, because it has much more relevance to those ones.

In principle I have the following major objections currently:

  • From my point of view the size of the particles is excessive for the kind of measurements applied. With particles of high density (btw. please specify the density) and particles sizes of 4 µm the solid is subject to sedimentation which falsifies the measurements.
  • The exclusion of CO2 in the electrokinetic measurements is not stated. And it is not stated whether the oxide may have had some carbon dioxide adsorbed during the salt titrations. Since inorganic carbon may adsorb and influence the pH, this has to be avoided or one has to find arguments to exclude the possibility. In the present case probably the absence of an effect of phosphate can be used as such an argument for the electrokinetic data.
  • The salt addition study is incomplete insofar as the different salts have not been tested. For example the LiClO4 could have been checked.

The remainder of the review is structured around a scanned copy of the manuscript with handwritten suggestions for changes and numbered items #i, to which I will refer in the following.

#1 What can be said about the levels of the zeta-potentials. Do they change in the expected way with salt concentration? Do they compare to other solids? Is there an effect of the nature of the salt?

#2 Please specify the salt level and nature here as far as possible. if not possible please state this.

#3 low pH is misleading, should be pH < pzc and pH > pzc

#4 it seems the most successful model in predicting pzc has been left out. If the particles here are this big, one might try to find out what the crystal faces are and apply the MUSIC model to get a better estimate for the pzc.

#5 the equations used should be given

#6 I think in the titrations the number >> 100 m2/L cannot be used as such. In a titration the absolute value of the surface area is relevant. I learnt that about 10 m2 should be in a vessel to get reasonable results. With 110 m2/L I could titrate a 50 ml sample and have 5.5 m2. This would not be enough. For titration data to be well described, one needs the specific surface area, the mass of solid in the vessel, the volume of solution/suspension and the titrant concentration (apart from the other procedure details).

#7 Even for one pure substrate different samples can have different IEPs. This has been shown for single crystals of different cuts for a number of oxides. The MUSIC model is able to describe this issue.

#8 As pointed out in the major objections I do not agree with this. My experience is that settling set in with 1 µm particles.

#9 I think the author lumps together terms. Site-binding models may include diffuse layer or Stern layer models and would all be termed surface complexation models.

#10 What about pH/Eh curves for the rarely studied oxide and maybe also an aqueous speciation plot that shows what is known? The recent Ekberg/Brown book may give the most recent reviewed data.

#11 As far as I know on silica anion adsorption is also very weak.

#12 Why not study here the adsorption of Ba as well in batch experiments?

#13 The effect of adsorption of Ba and increasing ionic strength cannot be easily distinguished. Which is maybe what the authors want to say.

#14 This section is confusing. If the particle are of 4 µm size how come here we find below 1000 nm?

#15 Here I refer to the other major objection, namely the CO2 issue. Nitrogen is lighter than air, so CO2 is not so easily removed with nitrogen. How long was this initially equilibrated?

#16 Why not do similar experiments with LiClO4, where the most drastic effects have been observed.

#15 has to be stated what was done to avoid CO2! How was specific surface area measured (outgassing, which gas)? How was the pH measured and the set-up calibrated? What was the sources of the salts?

Overall there are too many problems to make this immediately publishable.

Author Response

This is an important paper dealing with surface properties of an oxide that has previously been little explored. I wonder why it is not submitted to a colloid/interface journal, because it has much more relevance to those ones.

this manuscript is submitted to a special issue „Interfacial Chemistry“ edited by Carlos Bravo Díaz and Sonia Losada-Barreiro. Molecules have many sections, also in physical chemistry

In principle I have the following major objections currently:

From my point of view the size of the particles is excessive for the kind of measurements applied. With particles of high density (btw. please specify the density)

the following phrase was added.

The specific density of TeO2 is 5.67 g/cm3.

and particles sizes of 4 µm the solid is subject to sedimentation which falsifies the measurements.

We agree that the figure of 4 µm reported in the original text might be misleading . We added the following phrase.as an explanation.

Figure 1 b shows the microscopic picture of the powder. Apparently the size distribution is very broad and we find particles substantially greater than 4 µm, but also sufficient number of small particles to carry out an electrophoretic measurement. We also believe that the particle radii obtained by DLS (vide infra) which were 200-400 nm when the zeta potential was >50 mV in absolute value represent the actual sizes of particles whose zeta potentials were measured while the larger particles settled down, and they were not transferred into the zetameter cell.

The exclusion of CO2 in the electrokinetic measurements is not stated.

We added the following phrase

While in our previous electrokinetic study with Fe(OH)2 we designed a special procedure to exclude O2 and CO2 and in our previous electrikinetic study with BeO we designed a special procedure to exclude CO2 , no special efforts were made to exclude O2 or CO2 in the current  electrokinetic study with TeO2. In acidic solution, the solubility of CO2 is low. Moreover we are not aware of any special affinity of TeO2 to O2 or CO2. Exclusion of O2 or CO2 is not necessary in electrokinetic studies of most metal oxides, and the IEP obtained  with and without exclusion of O2 or CO2  are often identical[6,7].

 And it is not stated whether the oxide may have had some carbon dioxide adsorbed during the salt titrations.

We added the following phrase

The initial dispersion was stirred in a stream of nitrogen for about 1 h before the titration to remove any trace of CO2 and O2 from the dispersion, and an overpressure prevented the flow of external air into the reactor. The salt titration relies on very small changes in pH (cf. Fig. 11) so any  trace of CO2 in the reactor may affect the results, especially at pH about 7. In this respect the salt titration is very different from electrophoresis where 0.01 pH unit or so is immaterial (cf. the sizes of the symbols in Fig. 2-10 which are about 0.1 pH unit).

Since inorganic carbon may adsorb and influence the pH, this has to be avoided or one has to find arguments to exclude the possibility. In the present case probably the absence of an effect of phosphate can be used as such an argument for the electrokinetic data.

The salt addition study is incomplete insofar as the different salts have not been tested. For example the LiClO4 could have been checked.

The following phrase was added.

The results presented in Fig. 11 are an example of an (expected) failure: the apparent PZC was dependent on the solid-to-liquid ratio. This is why we only performed such measurements with one salt (KCl, different  solid-to-liquid ratios) and we have not attempted similar measurements with other salts. Moreover the salt-specific surface-charging behavior is observed at high salt concentrations [14] while the salt titration is only efficient at low total salt concentration (addition of salt to dispersions, in which the salt concentration is already high has rather insignificant effect on pH, even far from the PZC).

The remainder of the review is structured around a scanned copy of the manuscript with handwritten suggestions for changes and numbered items #i, to which I will refer in the following.

#1 What can be said about the levels of the zeta-potentials. Do they change in the expected way with salt concentration? Do they compare to other solids? Is there an effect of the nature of the salt?

This is discussed in lines 114 to 119 of the original manuscript.

The following phrase was added.

The absolute value of the zeta potential at given pH decreases as the salt concentration increases according to a commonly observed trend. There was no specific salt effect (NaCl vs. KCl, etc.) at salt concentrations of 0.001 and 0.01 M.

#2 Please specify the salt level and nature here as far as possible. if not possible please state this.

The following phrase was added.

(ionic strength not specified)

#3 low pH is misleading, should be pH < pzc and pH > pzc

low pH was replaced by below PZC

 high pH was replaced by above PZC

#4 it seems the most successful model in predicting pzc has been left out. If the particles here are this big, one might try to find out what the crystal faces are and apply the MUSIC model to get a better estimate for the pzc.

This can be done, but this is enough work  for a separate full length article (rather than correction in a communication)

#5 the equations used should be given

The following phrase was added.

PZC= 14.9 -2.19 valence ·  (not weighted)  (3)

PZC= 15- 2.26 valence ·  ( weighted)  (4)

PZC= 17.74 - 1.36 · electronegativity of oxide ·  ( not weighted)  (5)

PZC= 17.88 -1.4 · electronegativity of oxide ·  ( weighted)  (6)

PZC= 12.4-0.88 · z/r ·  ( not weighted)  (7)

PZC= 11.66 -0.75 · z/r ·  (  weighted)  (8)

PZC= 15.03-7.73 · z/R ·  ( not weighted)  (9)

PZC= 15.28-8.08 · z/R ·  (  weighted)  (10)

#6 I think in the titrations the number >> 100 m2/L cannot be used as such. In a titration the absolute value of the surface area is relevant. I learnt that about 10 m2 should be in a vessel to get reasonable results. With 110 m2/L I could titrate a 50 ml sample and have 5.5 m2. This would not be enough. For titration data to be well described, one needs the specific surface area, the mass of solid in the vessel, the volume of solution/suspension and the titrant concentration (apart from the other procedure details).

First of all our >> is „much greater“ not just „greater“. Then 110 is not >> 100.  We added the following phrase to explain our point.

The condition recalled above (>> 100 m2/L) is a rule of thumb rather than a strict principle, and the applicability of salt titration depends also on other factors than the solid-to-liquid ratio.

#7 Even for one pure substrate different samples can have different IEPs. This has been shown for single crystals of different cuts for a number of oxides. The MUSIC model is able to describe this issue.

Correct, but we deal with a powder here. We do not like to discuss a topic (single crystals) which is not directly related to our findings.

#8 As pointed out in the major objections I do not agree with this. My experience is that settling set in with 1 µm particles.

We added a long explanation:

Figure 1 b shows the microscopic picture of the powder. Apparently the size distribution is very broad and we find particles substantially greater than 4 µm, but also sufficient number of small particles to carry out an electrophoretic measurement. We also believe that the particle radii obtained by DLS (vide infra) which were 200-400 nm when the zeta potential was >50 mV in absolute value represent the actual sizes of particles whose zeta potentials were measured while the larger particles settled down, and they were not transferred into the zetameter cell.

#9 I think the author lumps together terms. Site-binding models may include diffuse layer or Stern layer models and would all be termed surface complexation models.

The text in patentheses is not directly related to our findings and it was deleted.

#10 What about pH/Eh curves for the rarely studied oxide and maybe also an aqueous speciation plot that shows what is known? The recent Ekberg/Brown book may give the most recent reviewed data.

Following the suggestion of the referee we added section 2.4. However instead of the book (nothing about tellurium is there) we refer to a journal paper

2.4 Solubility

In older literature (cf. ref. 18 for details) we fiund allegations that PZC of metal oxides matches the pH of minimum solubility. The present authors do not recommend pH of minimum solubility as a method of determination of PZC, but we can easily check how it works with TeO2, because the solubility data are available [21].

[21] Murase, K.; Suzuki, T.; Umenaka, Y.; Hirato, T.; Awakura, Y. Thermodynamics of Cathodic ZnTe Electrodeposition Using Basic Ammoniacal Electrolytes: Why CdTe Can Deposit While ZnTe Cannot. High Temp. Mater. Proc.,2011, 30, 451–458,  DOI 10.1515/HTMP.2011.068

According to Fig. 1 in [21] the  minimum of solubility falls at pH 5.3, which coincides with the lowest predicted PZC in Table 1. The minimum  solublility is 10-7 M (total concentration of soluble Te (IV) species), and it increases to 10-4 M at pH 2, and to 10-1 M at pH 10. Ref. [21] presents also a Pourbaix-type plot for Te. Within the electrochemical window of water elementary Te, Te(IV) and Te(VI) can be thermodynamically stable over various ranges of the redox potential.

#11 As far as I know on silica anion adsorption is also very weak.

Correct.The following phrase was added.

In contrast with metal oxides, the effect of inorganic anions on the zeta potential of silica was seldom studied. A few results compiled in ref. [18] show that the uptake of  inorganic anions by silica is low, and their effect on its  the zeta potential is rather insignificant.

#12 Why not study here the adsorption of Ba as well in batch experiments?

This is because we do not expect any substantial uptake of Ba from solution. First the specific surface area is too low. Moreover insignificant effect of Ba on zeta potential suggests low uptake.

#13 The effect of adsorption of Ba and increasing ionic strength cannot be easily distinguished. Which is maybe what the authors want to say.

the text was reworded.

Depression of negative zeta potentials in 10-3 M Ba(NO3)2 at pH>7 with respect to 10-3 M NaCl can be interpreted as the effect of increase in the ionic strength and it does not unequivocally indicate specific adsorption of Ba.

#14 This section is confusing. If the particle are of 4 µm size how come here we find below 1000 nm?

We added a long explanation:

Figure 1 b shows the microscopic picture of the powder. Apparently the size distribution is very broad and we find particles substantially greater than 4 µm, but also sufficient number of small particles to carry out an electrophoretic measurement. We also believe that the particle radii obtained by DLS (vide infra) which were 200-400 nm when the zeta potential was >50 mV in absolute value represent the actual sizes of particles whose zeta potentials were measured while the larger particles settled down, and they were not transferred into the zetameter cell.

#15 Here I refer to the other major objection, namely the CO2 issue. Nitrogen is lighter than air, so CO2 is not so easily removed with nitrogen. How long was this initially equilibrated?

Lighter, but first of all miscible, especially in turbulent flow.

The following phrase was added.

The initial dispersion was stirred in a stream of nitrogen for about 1 h before the titration to remove any trace of CO2 and O2 from the dispersion, and an overpressure prevented the flow of external air into the reactor. The salt titration relies on very small changes in pH (cf. Fig. 11) so any  trace of CO2 in the reactor may affect the results, especially at pH about 7. In this respect the salt titration is very different from electrophoresis where 0.01 pH unit or so is immaterial (cf. the sizes of the symbols in Fig. 2-10 which are about 0.1 pH unit).

#16 Why not do similar experiments with LiClO4, where the most drastic effects have been observed.

The following phrase was added.

The results presented in Fig. 11 are an example of an (expected) failure: the apparent PZC was dependent on the solid-to-liquid ratio. This is why we only performed such measurements with one salt (KCl, different  solid-to-liquid ratios) and we have not attempted similar measurements with other salts. Moreover the salt-specific surface-charging behavior is observed at high salt concentrations [14] while the salt titration is only efficient at low total salt concentration (addition of salt to dispersions, in which the salt concentration is already high has rather insignificant effect on pH).

#15 has to be stated what was done to avoid CO2! How was specific surface area measured (outgassing, which gas)?

The following phrase was added.

Adsorption of nitrogen at its boiling point was studied. In view of low specific surface area we used relatively large samples of powder (about 1 g) in specific surface area measurements, and we repeated the measurement 3 times. The samples were outgased at 300 oC for 1 h. The data points for p/p0<0.3 were used  to calculate .the specific surface area from the linearized form of BET equation.

How was the pH measured and the set-up calibrated?

The following phrase was added.

The pH measurements were carried out using a combined electrode calibrated against 3 commercial, fresh pH-buffers (pH 4, 7 and 10, POCh, Lublin, Poland). The pH was measured just before the injection of the dispersion into the zetameter cell. The time of the contact between the electrode (glass!) and the dispersion was minimized to avoid silica contamination.

What was the sources of the salts?

  The salts were from POCh Lublin Poland, as stated in lines 243-244 of the original text

Overall there are too many problems to make this immediately publishable.

Round 2

Reviewer 2 Report

I have commented on the answers to reviewers. It is included in red in the attached document. I recommend minor revisions/clarifications. 

Author Response

We only refer to the points which were still controversial.

While in our previous electrokinetic study with Fe(OH)2 we designed a special procedure to exclude

O2 and CO2 and in our previous electrikinetic study with BeO we designed a special procedure to

exclude CO2 , no special efforts were made to exclude O2 or CO2 in the current electrokinetic study

with TeO2. In acidic solution, the solubility of CO2 is low. Moreover we are not aware of any special

affinity of TeO2 to O2 or CO2. Exclusion of O2 or CO2 is not necessary in electrokinetic studies of

most metal oxides, and the IEP obtained with and without exclusion of O2 or CO2 are often

identical[6,7].

2

I do not agree here. In cases where the carbonate adsorbs, the IEP is affected, and since you do not

know whether it adsorbs, it is inconclusive. An acceptable reasoning would be to say that little

interaction with negatively charged surfaces is expected for CO2 (and cite e.g. work on silica), but that further clarification will be required. Another check could be the occurrence of aqueous complexes between carbonate and Te(IV). If the do not exist (or have not been found yet) this could serve as a support. Both issues should be coverd.

We added the following.

We also conducted a literature search on possible carbonate complexes of inorganic tellurium, but apparently such complexes do not exist or at least have not been detected yet. Negative charge of TeO2 particles is also an argument against hypothetical adsorption of carbonate anions and their hypothetical effect on the zeta potential.

#4 it seems the most successful model in predicting pzc has been left out. If the particles here are this

big, one might try to find out what the crystal faces are and apply the MUSIC model to get a better

estimate for the pzc.

This can be done, but this is enough work for a separate full length article (rather than correction in a communication)

I tend to disagree here as well. The original MUSIC model for example could be quickly used. The

second version could also be used rather quickly to just get an estimate. And such estimates are

typically much better than the ones used in the current version of the manuscript.

We added the following.

The MUSIC model [22] offers a possibility of prediction of PZC of oxides from the crystallographic data. So far it has been used for oxides whose PZC is already well-documented, but it can also be used for oxides whose PZC is less well-documented.

First of all our >> is „much greater“ not just „greater“. Then 110 is not >> 100. We added the following

phrase to explain our point.

The condition recalled above (>> 100 m2/L) is a rule of thumb rather than a strict principle, and the

applicability of salt titration depends also on other factors than the solid-to-liquid ratio.

Again I tend to disagree. I can use 600 m2/L and titrate 1 ml (assuming I use a micropipette). I would

not get good data. So the total amount of surface is relevant not the surface area concentration. The

value of the titrated volume would be sufficient to clarify the situation.

The rule of thumb refers to normal experimental conditions (volume on the order of 100 mL). We agree that titration of 1 mL will not produce good data, but this is too obvious to explain this to our readers. The details are explained in the cited literature, and we do not have to repeat everything here.

#7 Even for one pure substrate different samples can have different IEPs. This has been shown for

single crystals of different cuts for a number of oxides. The MUSIC model is able to describe this

issue.

Correct, but we deal with a powder here. We do not like to discuss a topic (single crystals) which is not directly related to our findings.

As pointed out above, the simple MUSIC version (1989) would be able to give some quick information, and even the second one can be quickly used in an excel spreadsheet. All you need toget is some structural data. Of course you cannot get the crystal planes here that easily, but you can try at least to get the information available in the literature. Like most stable crystal planes etc. If it is not available then you can state that and say that the MUSIC model (1996) coiuld not be used for that reason.

We added the following.

The MUSIC model [22] offers a possibility of prediction of PZC of oxides from the crystallographic data. So far it has been used for oxides whose PZC is already well-documented, but it can also be used for oxides whose PZC is less well-documented.

#12 Why not study here the adsorption of Ba as well in batch experiments?

This is because we do not expect any substantial uptake of Ba from solution. First the specific surface

area is too low. Moreover insignificant effect of Ba on zeta potential suggests low uptake.

Disagree. It has nothing to do with what you expect. Ba also adsorbs to silica for example. It is just a

matter of the pH. If it is non-specifically bonding it will happen somewhere above the IEP. It should be feasible.

Not with a SSA of a fraction of one m2/g.